# A Novel Deep Learning Approach for Deepfake Image Detection

Ali Raza [1,*] , Kashif Munir [2,*] and Mubarak Almutairi [3,*]

1   Institute of Computer Science, Khwaja Fareed University of Engineering and Information Technology, Rahim Yar Khan 64200, Pakistan
2   Institute of Information Technology, Khawaja Fareed University of Engineering & IT, Rahim Yar Khan 64200, Pakistan
3   College of Computer Science and Engineering, University of Hafr Al Batin, Hafr Alabtin 31991, Saudi Arabia
*   Correspondence: cosc211501009@kfueit.edu.pk (A.R.); kashif.munir@kfueit.edu.pk (K.M.); mutairims@gmail.com (M.A.)

**Abstract:** Deepfake is utilized in synthetic media to generate fake visual and audio content based on a person's existing media. The deepfake replaces a person's face and voice with fake media to make it realistic-looking. Fake media content generation is unethical and a threat to the community. Nowadays, deepfakes are highly misused in cybercrimes for identity theft, cyber extortion, fake news, financial fraud, celebrity fake obscenity videos for blackmailing, and many more. According to a recent Sensity report, over 96% of the deepfakes are of obscene content, with most victims being from the United Kingdom, United States, Canada, India, and South Korea. In 2019, cybercriminals generated fake audio content of a chief executive officer to call his organization and ask them to transfer $243,000 to their bank account. Deepfake crimes are rising daily. Deepfake media detection is a big challenge and has high demand in digital forensics. An advanced research approach must be built to protect the victims from blackmailing by detecting deepfake content. The primary aim of our research study is to detect deepfake media using an efficient framework. A novel deepfake predictor (DFP) approach based on a hybrid of VGG16 and convolutional neural network architecture is proposed in this study. The deepfake dataset based on real and fake faces is utilized for building neural network techniques. The Xception, NAS-Net, Mobile Net, and VGG16 are the transfer learning techniques employed in comparison. The proposed DFP approach achieved 95% precision and 94% accuracy for deepfake detection. Our novel proposed DFP approach outperformed transfer learning techniques and other state-of-the-art studies. Our novel research approach helps cybersecurity professionals overcome deepfake-related cybercrimes by accurately detecting the deepfake content and saving the deepfake victims from blackmailing.

**Keywords:** artificial intelligence; cybersecurity; cybercrimes; deepfakes; deepfake detection; deep learning; transfer learning

## 1. Introduction

Deepfake is a category of synthetic media [1] in which fake content is generated based on existing content, usually a person's media. In 2017, the "deepfake" term was first invented by a Reddit user of the name deepfake. The fake content is based on video graphics, audio signals, and face swapping technology. The artificial intelligence-based general adversarial networks [2] are utilized for deepfake generation. The deepfakes involved in cybercrimes [3] include identity theft, cyber extortion, imposter scam, fake news, incite violence, financial fraud, cyberbullying, celebrity fake obscenity videos [4] for blackmailing, democratic election, and many more. Detecting deepfake media is a big challenge with high demand in digital forensics.

Deepfake is mainly used to threaten organizations and individuals. The cybercrimes related to deepfake are rising day by day. The creation of deepfakes is unethical and is a

severe crime. Over 96% of the deepfakes are of obscene content, according to a Sensity report [5]. The United Kingdom, United States, Canada, India, and South Korea are the victims of deepfakes. In 2019, cybercriminals scammed a chief executive officer through fake audio content by transferring $243,000 to their bank account [6]. Deepfake-related rising crimes must be controlled and detected with an advanced tool.

Face recognition technology is utilized to verify an individual's identity by using a face database [7]. Face recognition is a category of biometric security. Face recognition systems are primarily used in law enforcement and security to control cybercrimes. The two- and three-dimensional pixels-based face images are utilized for face detection. The critical patterns to distinguishing faces are based on the chin, ears, contour of the lips, shape of cheekbones, depth of eye sockets, and distance between eyes.

The face recognition technology utilizes deep learning-based networks [8] to identify and learn specific face patterns. The face-related data is converted into a mathematical representation. The deep learning methods are best known for representation learning. Deep neural networks consist of multiple layers of connected neurons. The deep learning methods learned prominent facial biometric patterns [9] for face recognition with high accuracy. The databases containing the faces are utilized for training deep learning-based models. The deep learning-based models outperform the face recognition capabilities of humans.

The primary study contributions for deepfake detection are analyzed in detail. Transfer learning and deep learning-based neural network techniques are employed in comparison. The Xception, NAS-Net, Mobile Net, and VGG16 are the employed transfer learning techniques. A novel DFP approach based on a hybrid of VGG16 and convolutional neural network architecture is proposed for deepfake detection. The novel proposed DFP outperformed other state-of-the-art studies and transfer learning techniques. Hyperparameter tuning is applied to determine the best-fit parameters for the neural network techniques to achieve maximum performance accuracy scores. The training and validation effects of employed neural network techniques are examined in time-series graphs. The confusion matrix analysis of employed neural network techniques is conducted for performance metrics validation.

The remainder of the study organization is as follows: the related literature studies to deepfakes are examined in Section 2. The study methodology is based on the research working flow analyzed in Section 3. The utilized deepfake dataset is described in Section 4. The transfer learning-based neural network techniques are examined in Section 5. The novel proposed deep learning-based approach is described in Section 6. The hyperparameter tuning of employed neural network techniques is conducted in Section 7. The discussions of results and scientific validations of our research study are evaluated in Section 8. Our novel research study is concluded in Section 9.

## 2. Related Literature

The deepfake-related literature studies are examined in this section. The recent state-of-the-art studies applied for deepfake detection are described. The literature analysis is based on the dataset utilized and performance accuracy scores. The deepfake detection-related literature summary is analyzed in Table 1.

Video deepfake detection using a deep learning-based methodology was proposed in this study [10]. The XGBoost approach was the proposed approach. The face areas were extracted from video frames using a YOLO face detector [11], CNN, and Inception Res-Net techniques. The CelebDF and FaceForencics++ datasets were utilized for model building and training. The extracted features from faces were input to XGBoost for detection. The proposed approach achieved a 90% accuracy score for video deepfake detection.

Automatic deepfake video classification using deep learning was proposed in this study [12]. The Mobile Net and Xception are the applied deep learning-based techniques. The FaceForensics++ dataset was utilized for deep learning model training and testing.

The applied deep learning technique's accuracy scores vary between 91% and 98% for deepfake video classification.

Deepfake recognition based on human eye blinking patterns using deep learning was proposed [13]. DeepVision was the proposed approach for deepfake detection. The static deepfakes eye blinking images dataset based on eight video frames was utilized for proposed model testing and training. The proposed model achieved an 87% accuracy score for deepfake detection.

The deepfake detection based on spectral, spatial, and temporal inconsistencies using multimodal deep learning techniques was proposed in this study [14]. The Facebook deepfake challenge dataset was utilized for learning model building. The multimodal network was proposed based on Long Short-Term Memory (LSTM) networks. The proposed model achieved a 61% accuracy score for deepfake detection.

**Table 1.** The deepfake detection-related literature summary and comparative analysis.

| Ref. | Year | Approach | Dataset | Accuracy Score (%) | Research Aim |
|---|---|---|---|---|---|
| [10] | 2021 | XGBoost | CelebDF and FaceForencics++ | 90 | The video deepfake detection using a deep learning-based methodology was proposed. |
| [12] | 2020 | Mobile Net and Xception | FaceForencics++ | 91 | Automatic deepfake video classification using deep learning was proposed. |
| [13] | 2020 | DeepVision | Static deepfakes eye blinking images dataset | 87 | The deepfake recognition based on a human eye blinking pattern using deep learning was proposed. |
| [14] | 2020 | Multimodal network | Facebook deepfake challenge dataset | 61 | The deepfake detection based on spectral, spatial, and temporal inconsistencies using multimodal deep learning techniques was proposed. |
| [15] | 2019 | Dense Net and fake feature network | CelebA | 90 | Deepfake image detection using pairwise deep learning was proposed. |
| [16] | 2022 | Dense Net | annotated CT-GAN | 80 | Medical deepfake image detection based on machine learning and deep learning was proposed. |
| **Proposed** | **2022** | **Novel DFP** | **Photoshopped real and fake faces** | **94** | **A novel DFP approach based on a hybrid of VGG16 and convolutional neural network architecture is proposed for deepfake image detection.** |

Deepfake image detection using pairwise deep learning was proposed in this study [15]. All research experiments were carried out by using the CelebA dataset. Generative adversarial networks generated the fake and real image pairs. The Dense Net and fake feature network were proposed for deepfake image detection. The proposed approach achieved a 90% accuracy score for deepfake detection.

The deepfake detection from manipulated videos using ensemble-based learning approaches was proposed in this study [17]. The DeepfakeStack was the proposed approach for deepfake detection. The DeepfakeStack approach training and testing were performed using the FaceForensics++ dataset.

Medical deepfake image detection based on machine learning and deep learning was proposed in this study [16]. The annotated CT-GAN dataset was employed for building learning techniques. The proposed Dense Net achieved an accuracy score of 80% for multiclass delocalized medical deepfake images.

The literature review demonstrates that no previous research study uses the optimized hybrid model architecture compared to our proposed study. Our proposed study achieved high metrics scores for deepfake detection compared to previous studies. The key findings of our study are based on a novel optimized hybrid model for deepfake detection. The model architecture utilized in our research study contains less complexity. The pro-

posed optimized model is efficient in the data processing. In conclusion, our novel research study has many valuable applications in cybersecurity for deepfake detection.

## 3. Study Methodology

The research methodology architectural analysis is examined in Figure 1. The deepfake images based on fake and real human faces are utilized for building employed neural network techniques. The fake and real faces are structured with the target label into a dataset. The structured deepfake dataset is split into train, validation, and test data portions. The 90% train portion of the dataset is utilized for training employed neural network techniques. The outperformed novel DFP approach is fully hyper-parametrized to give the best accuracy score in deepfake detection. The performance evaluation of neural network techniques on unseen test data is determined by 10% of the test portion. The novel proposed approach makes predictions on unseen data with high accuracy results. An advanced proposed deep learning-based approach is in a generalized form and ready to detect the fake and real faces in deployment.

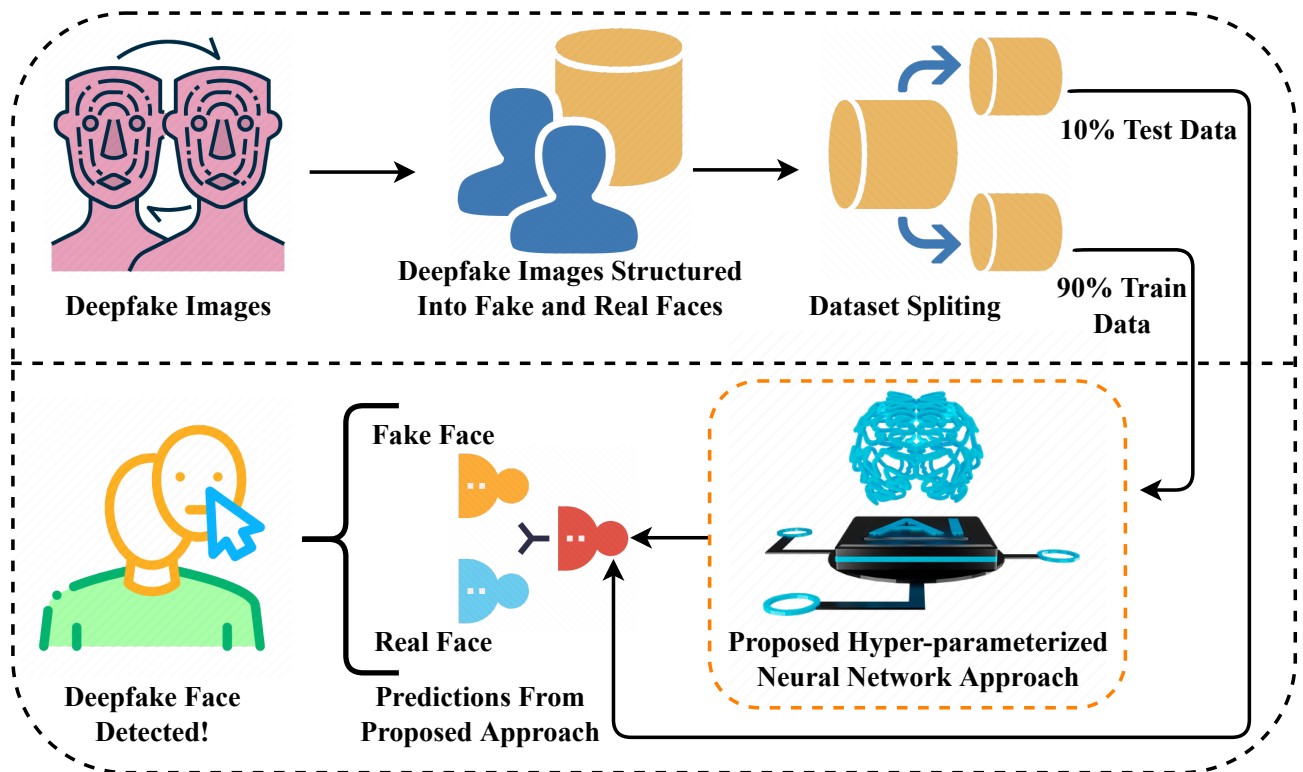

**Figure 1.** The methodological architectural analysis of our novel proposed research study in deepfake prediction.

## 4. Deepfake Dataset

The deepfake dataset was used for training and testing the employed neural network techniques. The benchmark deepfake dataset is publicly available on Kaggle by the Department of Computer Science, Yonsei University [18]. The deepfake dataset contains expert-generated photoshopped face images. The generated deepfake images combine numerous faces, separated by nose, eyes, mouth, and whole face. The dataset contains 1081 images of real and 960 images of fake faces. The sample images from the deepfake dataset are analyzed with the target label in Figure 2.

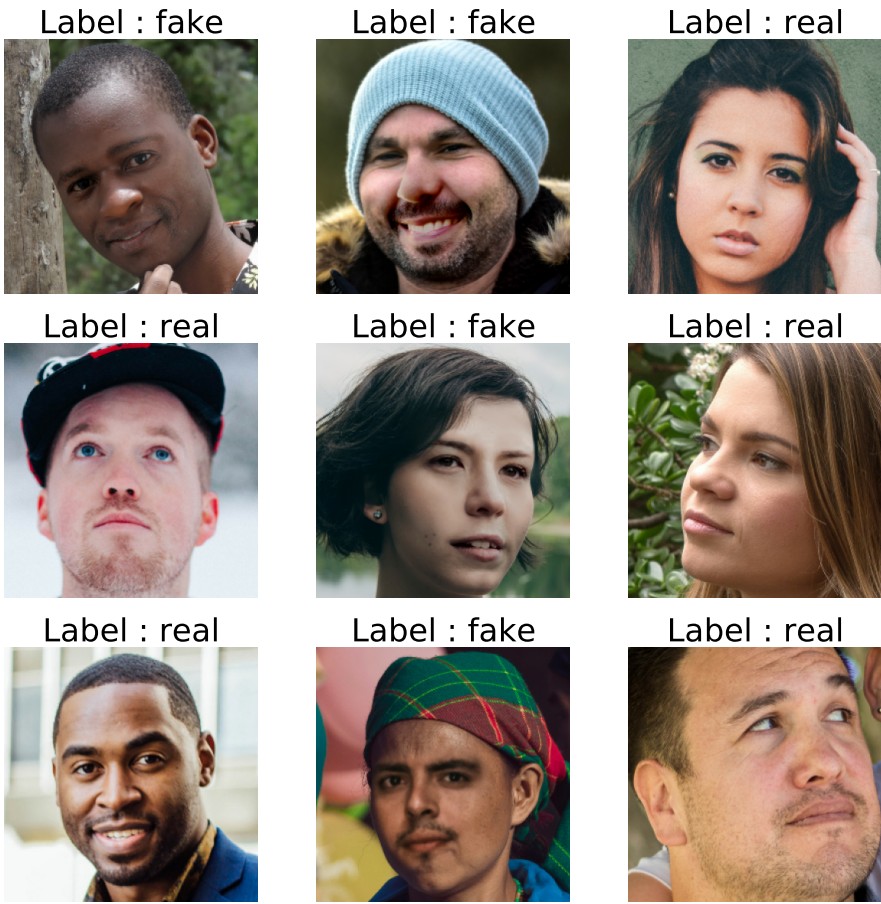

**Figure 2.** The deepfake dataset sample image analysis with the assigned label.

## 5. Applied Transfer Learning Neural Network Techniques

The employed transfer learning-based neural network techniques are examined in this section. Transfer learning [19] is an approach where we build pre-trained models for the prediction process. Transfer learning provides the ability to utilize the learned features for high performance in prediction. The transfer learning-based fine-tuning methods utilizes an already-trained network and re-trains part of the network on the new dataset. The working mechanism of transfer learning techniques utilized for deepfake detection is analyzed in this section. The configuration parameters and architectural analysis of neural network techniques are conducted.

### 5.1. Xception Technique

The Xception [20] is a transfer learning-based neural network technique primarily used for the image recognition task. The Xception model stands for extreme inception. The Xception model is the extension of the Inception architecture. The depth-wise separable convolution layers are used in the Xception model architecture. The Xception model has the smallest weight serialization. The 36 convolutional layers form the Xception model architecture.

The Xception model configuration parameters and layer architectural analysis are analyzed in Table 2. The analysis describes the first layer as the input layer with an output shape of (100, 256, 256, 3). Then, the Xception layers are involved with 2048 units and 20,861,480 total parameters. The architecture adds the dropout layer to prevent the model's overfitting. The flattened layers are involved in the architecture for converting the pixel's data into a series of one-dimensional arrays. The family of dense layers in the architecture are involved in deepfake prediction. The dense layer has 64 units with relu

activation. The deepfake classification is performed by the architecture's output layer with sigmoid activation.

**Table 2.** The Xception model configuration parameters and layer architectural analysis.

| Sr No. | Layers | Unit | Activation Function | Output Shape | Parameters |
|---|---|---|---|---|---|
| 1 | The Sequential Input Layers. | None | None | (100, 256, 256, 3) | 0 |
| 2 | The Xception Layers. | 2048 | None | (None, 8, 8, 2048) | 20,861,480 |
| 3 | The Dropout Layers. | 0.2 | None | (100, 8, 8, 2048) | 0 |
| 4 | The Flatten Layers. | None | None | (100, 131,072) | 0 |
| 5 | The Dense Layers. | 64 | RELU | (100, 64) | 8,388,672 |
| 6 | The Output Layers. | 1 | SIGMOID | (100, 1) | 65 |

*5.2. NAS-Net Technique*

The NAS-Net stands for neural search architecture (NAS) Network. The NAS-Net [21] is a transfer learning-based model belonging to the convolutional neural network family. The Google Brain reviews the NAS-Net. The NAS-Net model is trained on the ImageNet database [22], having more than a million images. The model has less computation costs. The blocks in the NAS-Net architecture are searched by the reinforcement learning search method. Our research study replaced the top layers of NAS-Net with MLP block for deepfake detection.

The NAS-Net model configuration parameters and layer architectural analysis are analyzed in Table 3. The analysis describes the first layer as the input layer with an output shape of (100, 331, 331, 3). Then, the NAS-Net layers are involved with 4032 units and 84,916,818 total parameters. The architecture adds the dropout layer to prevent the model's overfitting. The flattened layers and family of dense layers in the architecture are involved in deepfake prediction. The dense layer has 64 units with relu activation. The deepfake classification is performed by the architecture's output layer with sigmoid activation.

**Table 3.** The NAS-Net model configuration parameters and layer architectural analysis.

| Sr No. | Layers | Unit | Activation Function | Output Shape | Parameters |
|---|---|---|---|---|---|
| 1 | The Sequential Input layers. | None | None | (100, 331, 331, 3) | 0 |
| 2 | The NAS-Net layers. | 4032 | None | (None, 11, 11, 4032) | 84,916,818 |
| 3 | The Dropout layers. | 0.2 | None | (100, 11, 11, 4032) | 0 |
| 4 | The Flatten layers. | None | None | (100, 487,872) | 0 |
| 5 | The Dense layers. | 64 | RELU | (100, 64) | 31,223,872 |
| 6 | The Output layers. | 1 | SIGMOID | (100, 1) | 65 |

*5.3. Mobile Net Technique*

The Mobile Net [23] is a transfer learning model we have built for deepfake detection in our research study. Google open-sourced the Mobile Net model. The Mobile Net is highly used for vision-related applications with fast processing. The model architecture reduces computation costs and has less complexity. The architecture is built using depthwise separable convolutions [24]. The two operations, depthwise and pointwise, are performed in depthwise separable convolutions. This reduces the number of parameters compared to regular convolutions. Our research study replaced the top layers of Mobile Net with MLP block.

The Mobile Net model configuration parameters and layer architectural analysis are analyzed in Table 4. The analysis describes the first layer as the input layer with an output shape of (100, 256, 256, 3). Then, the Mobile Net layers are involved with 1024 units

and 3,228,864 total parameters. The parameters of Mobile Net are less as compared to other techniques. To prevent the overfitting of the model, the dropout layer is added. The deepfake prediction is performed by a family of dense and flattened layers in the architecture. The dense layer has 64 units with relu activation. The architecture's output layer with sigmoid activation is used for deepfake classification.

**Table 4.** The Mobile Net model configuration parameters and layer architectural analysis.

| Sr No. | Layers | Unit | Activation Function | Output Shape | Parameters |
|---|---|---|---|---|---|
| 1 | The Input Sequential layers. | None | None | (100, 256, 256, 3) | 0 |
| 2 | The Mobile Net layers. | 1024 | None | (None, 8, 8, 1024) | 3,228,864 |
| 3 | The dropout layers. | 0.2 | None | (100, 8, 8, 1024) | 0 |
| 4 | The Flatten layers. | None | None | (100, 65,536) | 0 |
| 5 | The Dense layers. | 64 | RELU | (100, 64) | 4,194,368 |
| 6 | The Output layers. | 1 | SIGMOID | (100, 1) | 65 |

### 5.4. VGG16 Technique

The VGG16 [25] is a pre-trained neural network technique primarily used for image recognition tasks. K. Simonyan and A proposed the VGG16 model. The VGG16 is based on the convolution neural net (CNN) architecture. In 2014, the VGG16 model architecture was first utilized in the ILSVR competition. We built the VGG16 model on our dataset for deepfake detection. The top layers of VGG16 are replaced by a multi-layer perceptron (MLP) [26] block in our research study.

The VGG16 model configuration parameters and layer architectural analysis are analyzed in Table 5. The analysis demonstrates that the model's first layer is the input layer with an output shape of (100, 256, 256, 3). Then, VGG16 model layers are involved with 512 units and a large number of training parameters of 14,714,688. A dropout layer is a subsequent layer in the architecture to prevent the overfitting of the model. Then, flattened layers are involved in converting the pixel data into a series of a one-dimensional array. A family of dense layers is involved for deepfake prediction. The dense layer has eight units with relu activation. The final output layer with sigmoid activation is used for deepfake classification.

**Table 5.** The VGG16 model configuration parameters and layer architectural analysis.

| Sr No. | Layers | Unit | Activation Function | Output Shape | Parameters |
|---|---|---|---|---|---|
| 1 | The Input Sequential layers. | None | None | (100, 256, 256, 3) | 0 |
| 2 | The VGG16 layers. | 512 | None | (None, 8, 8, 512) | 14,714,688 |
| 3 | The Dropout layers. | 0.2 | None | (100, 8, 8, 512) | 0 |
| 4 | The Flatten layers. | None | None | (100, 32,768) | 0 |
| 5 | The Dense layers. | 64 | RELU | (100, 64) | 2,097,216 |
| 6 | The Output layers. | 1 | SIGMOID | (100, 1) | 65 |

### 6. Novel Proposed Approach

A novel DFP approach is proposed based on a hybrid of VGG16 and convolutional neural network architecture. For the first time, our research study uses a hybrid of transfer learning and deep learning-based neural network architecture for deepfake detection. The VGG16 and convolutional neural network layers were combined to make the proposed model architecture. Convolutional neural networks are the family of artificial neural networks primarily used in image recognition applications. The convolutional neural networks are specifically designed to process pixel data. The image data is represented in

convolutional neural networks using multidimensional array forms. The primary aim of the artificial neural network is to learn patterns from historical data and make predictions for unseen data.

The proposed model layer's architecture and configuration parameters are analyzed. The configuration parameters of the novel proposed approach are examined in Table 6. The configuration parameters described the units and parameters used during the proposed model building. The architecture analysis is visualized in Figure 3. The architectural analysis demonstrates the deepfake detection image data flow from input to prediction layers using the proposed technique. The hybrid layers of VGG16 and convolutional neural networks are involved in creating the architecture. The pooling, dropout, flatten, and fully-connected layers were combined to build the novel proposed model architecture.

**Table 6.** The novel proposed DFP model configuration parameters analysis.

| Sr No. | Layers | Unit | Activation Function | Output Shape | Parameters |
|--------|--------|------|--------------------|--------------|------------|
| 1 | The Sequential Input layers. | None | None | (100, 256, 256, 3) | 0 |
| 2 | The VGG16 Network layers. | 512 | None | (None, 8, 8, 512) | 14,714,688 |
| 3 | The Convolutional Network layers. | 1025 | RELU | (100, 6, 6, 1025) | 4,724,225 |
| 3 | The max pooling layers. | $2 \times 2$ | None | (100, 3, 3, 1025) | 0 |
| 4 | The Dropout layers. | 0.02 | None | (100, 3, 3, 1025) | 0 |
| 5 | The Flatten layers. | None | None | (100, 9225) | 0 |
| 6 | The Dense layers. | 1025 | RELU | (100, 1025) | 9,456,650 |
| 7 | The Dense layers. | 1 | SIGMOID | (100, 1) | 1026 |

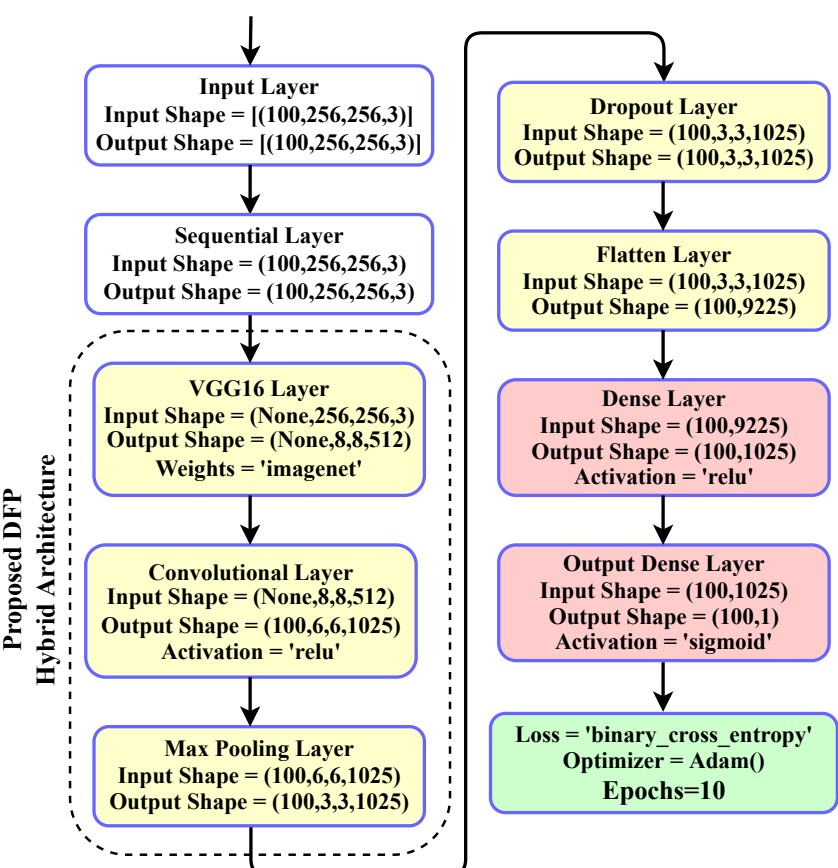

**Figure 3.** The novel proposed DFP model architectural analysis.

The novel proposed DFP model has many advantages compared to state-of-the-art models. The proposed DFP network architecture is fully optimized. The comparative architectural analysis demonstrates that the proposed model contains less complexity. The proposed optimized model is efficient in data processing. The high-performance metrics scores were obtained for deepfake detection using the proposed three stacks of convolutional neural network layers.

## 7. Hyperparameter Tuning of Employed Neural Network Techniques

Hyperparameter tuning is a recursive process of training and testing a neural network technique [27]. The hyperparameter tuning of employed neural network techniques was applied to determine the best-fit parameters for deepfake detection. The best-fit parameters resulted in higher performance accuracy results. The hyperparameters of employed neural network techniques are examined in Table 7. The optimizer was utilized to update weights in the neural network. The loss function controlled the loss of the neural network during training. The accuracy measure was used as the metric. The output layer used sigmoid activation. Verbose and epochs represent the neural network technique's data learning iterations.

**Table 7.** The best-fit hyperparameter tuning analysis of all employed neural network techniques.

| Hyperparameters | Values |
| --- | --- |
| Optimizer | Adam |
| loss Function | Binary Cross Entropy |
| Metrics | Accuracy |
| Activation | Sigmoid |
| Epochs | 20 |
| Verbose | 1 |

## 8. Results and Discussions

The research results are examined with detailed discussions in this section. The programming tools and scientific evaluation metrics for employed neural network techniques are examined.

### 8.1. Experimental Setup

The research experimental setup was utilized to build the neural network techniques. The Python programming language [28] was used for evaluating all research experiments. TensorFlow module with version 2.8.2 and Keras module with version 2.8.0 were utilized for building the employed neural network techniques. The scientific evaluations of the research study based on performance metrics were examined. The time computation, loss sore, accuracy score, precision score, f1 score, specificity score, and geometric mean score were performance metrics utilized for performance evaluations.

### 8.2. Results Performance Analysis of Employed Techniques

The time series-based training effects of employed techniques by epochs are examined in Figure 4. The analysis shows that the NAS-Net, Xception, and Mobile Net models achieved low-performance scores during training. From epochs one to three of the NAS-Net, Xception, and Mobile Net models, the training loss was near 1.4, which is comparatively high. The analysis shows that the VGG16 and DFP models achieved good scores in comparison. From epochs one to three of the VGG16 and DFP, the training loss was below 0.1, which shows good performance for deepfake detection. This analysis shows the loss score is decreased gradually with an increase in epochs.

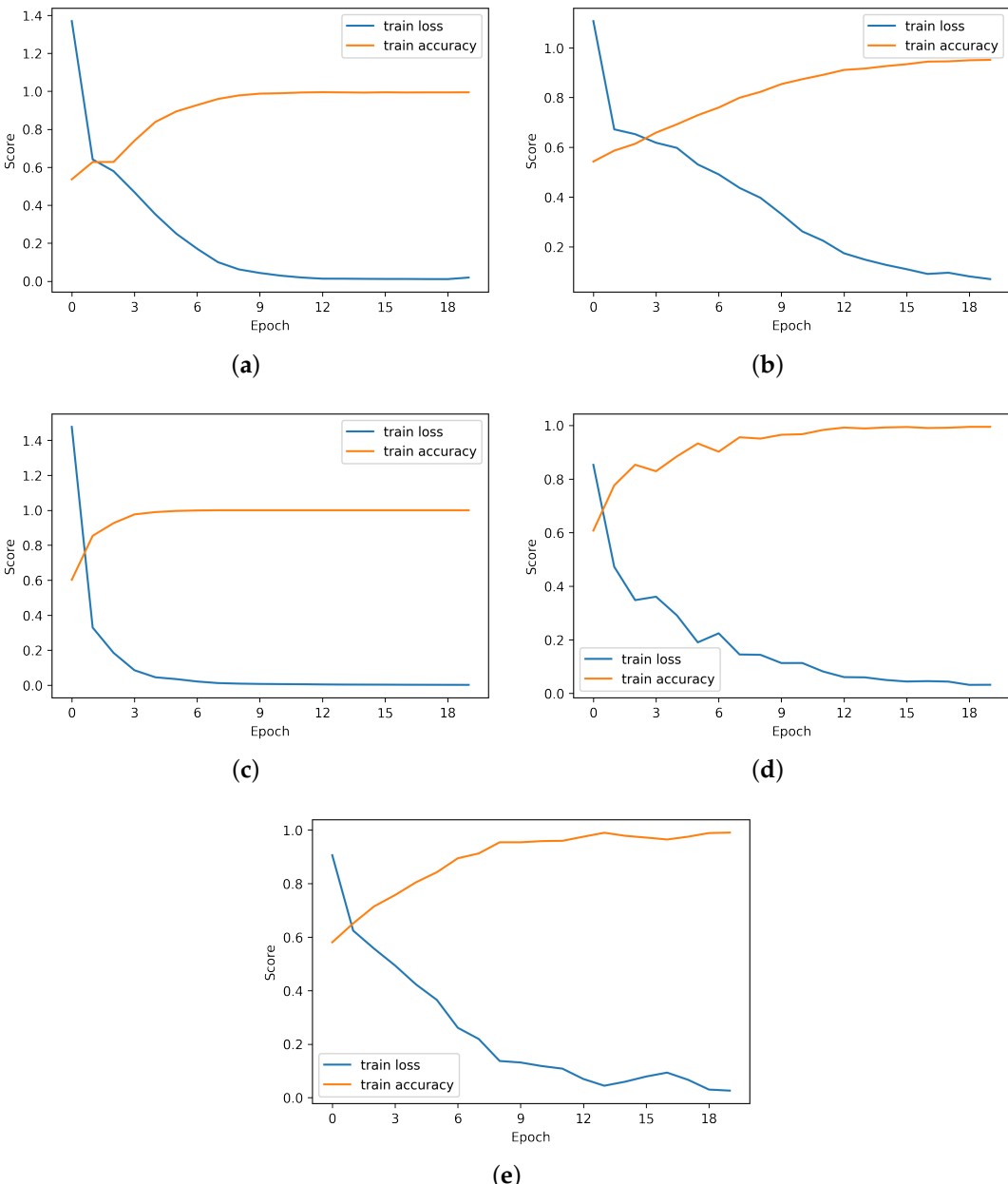

**Figure 4.** The time-series analysis of the employed neural network approaches with each epoch during training. (**a**) The loss and accuracy analysis of the NAS-Net approach. (**b**) The loss and accuracy analysis of the Xception approach. (**c**) The loss and accuracy analysis of the Mobile Net approach. (**d**) The loss and accuracy analysis of the VGG16 approach. (**e**) The loss and accuracy analysis of the proposed approach.

The comparative performance analysis of employed neural network techniques on unseen test data is analyzed in Table 8. The analysis demonstrates that the NAS-Net model achieved poor performance for all performance metrics. The NAS-Net model had an accuracy score of 83% and a high loss score. The Xception approach achieved a precision of 82% with a less loss score, which is good compared to the NAS-Net model. The Mobile Net technique achieved an accuracy of 88%, which is acceptable compared to Xception and NAS-Net models. The VGG16 model achieved favorable scores, with an accuracy score of 90% compared to Xception, Mobile Net, and NAS-Net models. The loss score of the VGG16 model was moderate. The analysis demonstrates that our proposed approach outperformed all performance metrics with an immense accuracy score of 94%. The loss

score was reduced to 0.2 by the proposed approach. The analysis shows the superiority of the novel proposed approach for performance metrics scores compared to other employed neural network techniques.

**Table 8.** The comparative performance analysis of employed neural network techniques on unseen test data.

| Technique | Accuracy Score (%) | Loss Score | Precision Score (%) | F1 Score (%) | Specificity Score (%) | Geometric Mean Score (%) |
|---|---|---|---|---|---|---|
| NAS-Net | 83 | 0.8 | 80 | 86 | 73 | 86 |
| Xception | 84 | 0.5 | 82 | 86 | 75 | 87 |
| Mobile Net | 88 | 0.4 | 86 | 89 | 84 | 88 |
| VGG16 | 90 | 0.4 | 88 | 92 | 84 | 91 |
| **Proposed DFP** | **94** | **0.2** | **95** | **94** | **94** | **94** |

The bar chart-based performance comparative analysis of employed neural network techniques is visualized in Figure 5. Our proposed model achieved 94% accuracy, f1, and geometric mean scores compared to other neural network techniques. The Xception and NAS-Net-models showed low-performance scores. The Mobile Net and VGG16 approaches achieved acceptable scores for all metrics but less than the proposed model with a huge difference. All employed learning techniques achieved fair geometric mean scores. The analysis demonstrates that our novel proposed approach achieved high scores for all performance metrics.

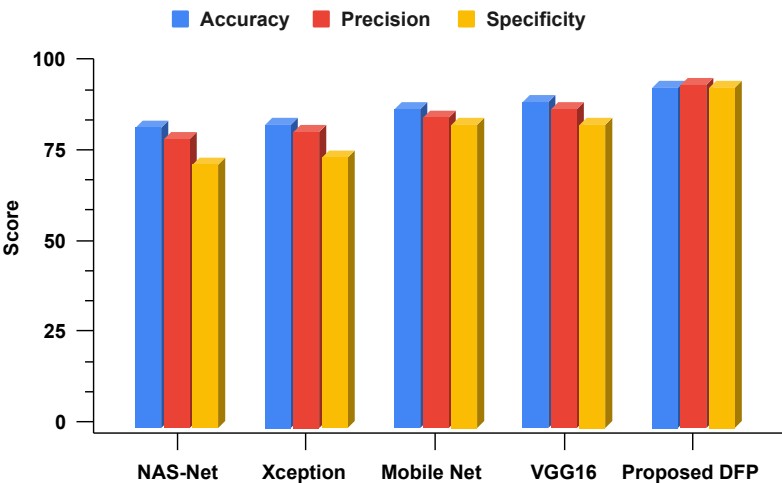

**Figure 5.** The bar chart-based performance comparative analysis of employed neural network techniques.

The error loss score-based bar chart analysis of employed neural network techniques is visualized in Figure 6. The proposed approach achieved a minimal loss score of 0.2, validating the high-performance accuracy for deepfake detection. The analysis shows that the highest loss score of 0.8 was achieved by the NAS-Net technique, followed by Xception, Mobile Net, and VGG16. The analysis demonstrates that our novel proposed approach achieved less loss scores compared to employed neural network techniques. This analysis provides the validity of our novel proposed model for high-performance metric scores.

The confusion matrix analysis for validating the performance metrics results of employed neural network techniques is examined in Figure 7. The high-class prediction error rate was achieved by the Xception model, followed by Mobile Net and NAS-Net. The confusion matrix analysis shows the overall prediction performance analysis using learning techniques. The novel proposed approach achieved a high confusion matrix analysis score. The analysis demonstrates that the proposed approach has a minimum error rate in class predictions compared to confusion matrix results.

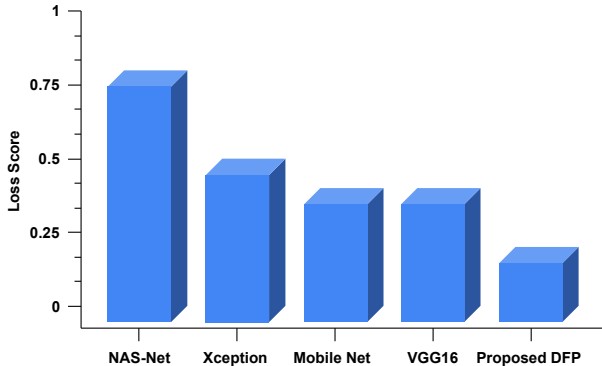

**Figure 6.** The bar chart-based loss score comparative analysis of employed neural network techniques.

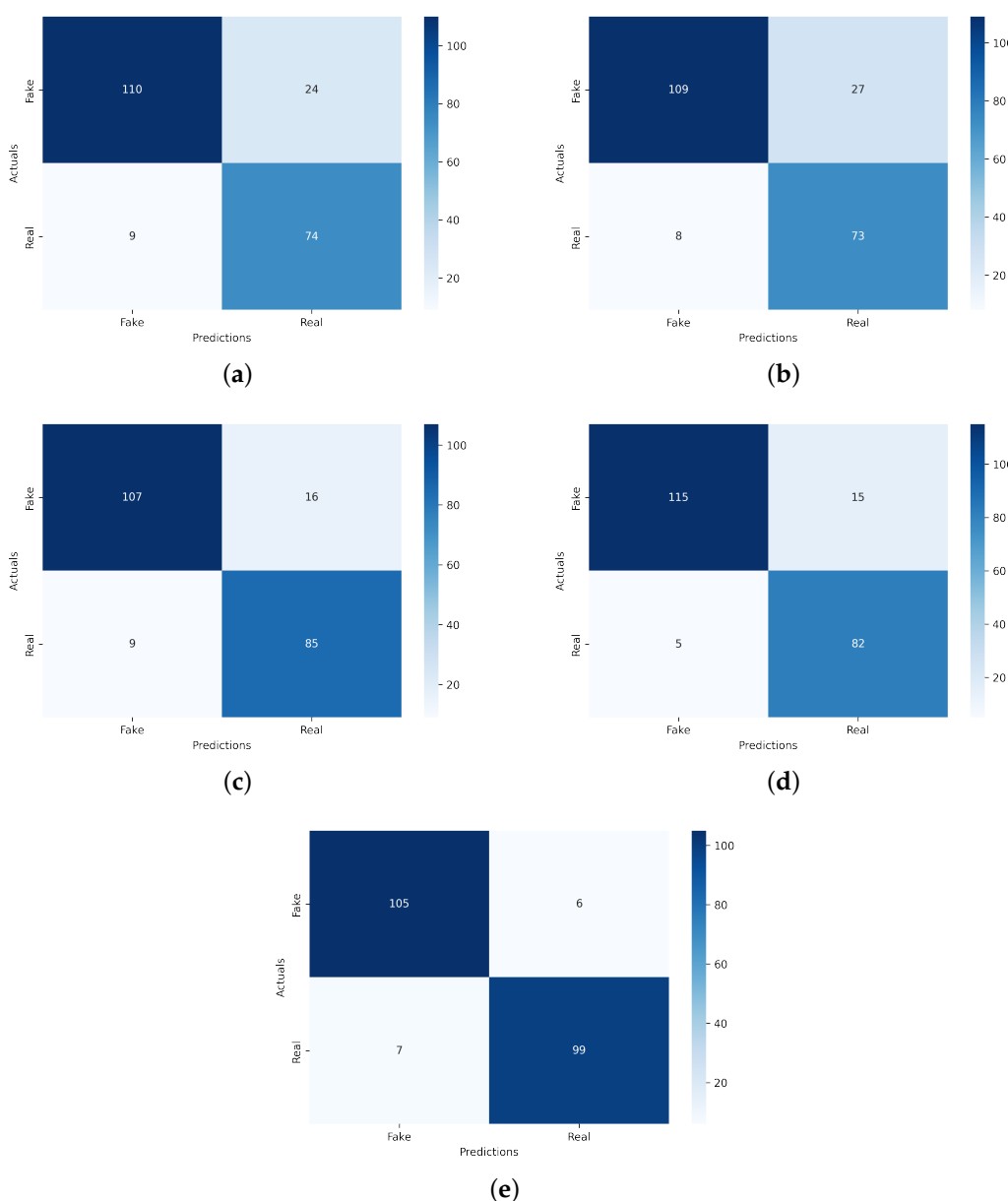

**Figure 7.** The confusion matrix analysis of employed neural network techniques is based on the validations of performance metrics results. (**a**) The confusion matrix of the Xception approach. (**b**) The confusion matrix of the NAS-Net approach. (**c**) The confusion matrix of the Mobile Net approach. (**d**) The confusion matrix of the VGG16 approach. (**e**) The confusion matrix of the proposed approach.

### 8.3. K-Fold Cross Validation Results

The performance validation of our novel proposed approach is analyzed in Table 9. The k-fold cross-validation technique based on four folds of data was used for proposed model validations. The analysis shows that the proposed model achieved an average accuracy score of 95% with a standard deviation of ±1. The cross-validation results demonstrate that our proposed model is in the generalized form to detect deepfakes from social media.

**Table 9.** The performance validation analysis of the novel proposed approach based on the k-fold cross-validation technique.

| K-Folds | Accuracy Score (%) | Loss Score |
|---|---|---|
| Fold 1 | 93 | 0.13 |
| Fold 2 | 95 | 0.08 |
| Fold 3 | 95 | 0.13 |
| Fold 4 | 96 | 0.08 |
| **Average** | **95** | **0.10** |
| **Standard Deviation** | **±1** | |

### 8.4. Comparison With State-of-the-Art Studies

The comparative performance analysis of our novel proposed DFP with the other state-of-the-art studies is examined in Table 10. The recent state-of-the-art approaches from 2020 and 2021 are taken for comparison. The deep learning and transfer learning-based approach were built in our research study. The traditional convolutional neural network, VGG16, and Mobile Net are the state-of-the-art approach we built and applied to our deepfake dataset. The build approaches on our employed dataset are examined in comparison with the proposed approach. The analysis demonstrates that our novel proposed approach outperformed the other state of art studies.

**Table 10.** The comparative performance analysis of the proposed approach with the other state of art studies.

| Ref. | Year | Learning Type | Technique | Accuracy Score (%) | Precision Score (%) |
|---|---|---|---|---|---|
| [29] | 2020 | Deep learning | Traditional Convolutional Neural Network | 89 | 89 |
| [30] | 2021 | Transfer learning | VGG16 | 90 | 88 |
| [31] | 2020 | Transfer learning | Mobile Net | 88 | 86 |
| **Proposed** | **2022** | **Deep learning** | **Novel DFP** | **94** | **95** |

## 9. Conclusions

Deepfake detection using a novel deep learning-based approach is proposed to help cybersecurity professionals overcome deepfake-related cybercrimes by accurately detecting the deepfake content. The Xception, NAS-Net, Mobile Net, and VGG16 are the employed neural network techniques in comparison. The benchmark deepfake dataset containing the real and fake faces was utilized for building all research models. The proposed DFP approach outperformed with employed learning techniques and state-of-the-art studios. The proposed model achieved 94% accuracy for deepfake detection. The employed neural network techniques were validated using confusion matrix analysis and analyzed through a time series analysis. In the future, blockchain-based technology and cloud web system will be designed to configure our proposed system for detecting the deepfakes through a secure online system.

**Author Contributions:** Conceptualization, methodology, software, validation, formal analysis, investigation, resources, data curation, writing, and original draft preparation, review and editing, and visualization, A.R.; supervision, project administration, and funding acquisition, K.M. and M.A. All authors have read and agreed to the published version of the manuscript.

**Funding:** This work was supported by University of Hafr Albatin, Saudi Arabia.

**Institutional Review Board Statement:** Not applicable.

**Informed Consent Statement:** Informed consent was obtained from all subjects involved in the study.

**Data Availability Statement:** The supporting data for the findings of this study are available from the corresponding author on reasonable request.

**Acknowledgments:** The authors would like to thank all participants for their fruitful cooperation and support.

**Conflicts of Interest:** The authors declare no conflict of interest.

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
