# Peer review of "A Novel Deep Learning Approach for Deepfake Image Detection"

_applsci, doi:10.3390/app12199820_

Round 1

Reviewer 1 Report (Previous Reviewer 1)

No further comments.

Author Response

Response to Reviewer 1 Comments

Manuscript ID: applsci-1930763

Title: A Novel Deep Learning Approach for Deepfake Image Detection

Authors: Ali Raza*, Kashif Munir*, Mubarak Almutairi*

Dear Editor,

Thank you very much for allowing a resubmission of our manuscript, with an opportunity to address the reviewers’ comments.

We would like to thank the editor and all the worthy reviewers for their efforts, valuable comments and suggestions. The comments given improved the quality of our paper. Based on the feedback, we have extensively revised our manuscript. The updated manuscript with red highlighting indicates changes. The detailed modifications to address reviewers’ comments are provided point-by-point in the following. For clarity, we have marked our responses in blue.

REVIEWER 1

Point: No further comments.

Response: The authors are highly grateful for your efforts and insightful comments. The authors appreciate your generous efforts towards the paper revision.

Reviewer 2 Report (New Reviewer)

This is an interesting area – and the overall approach is good. However, I think the presentation of the results could be improved, and I also think the actual analysis/experiment leaves room for improvement also – but perhaps some of this is from how it was presented in the paper.

54-64 feels clumsy – it is in two bullets – why? And then the contents don’t flow. I suggest just writing them as normal content that flows.

“Other studies have failed to achieve high metrics scores compared with our research study.”  This study is 94 – and there in the table has two 90s and a 93. Not convinced this is as significant a leap as you say to say that others have failed..

There is no mention of cross-validation – so with such a small dataset, and then testing with just one lot of 10%, I am not convinced that 94% is actually an accurate number. The 10% testing set could be a ‘lucky’ 10%.

“The François Chollet” – He is just a software engineer, not sure he warrants “The” as his title.

“The Python programming tool [28] is used for evaluating all research experiments.” ??? what tool? Python is a language – so you mean the tool that is at reference 28. State it in the text, don’t make the reader go to the references to then see that it points to a paper, that they then need to go lookup to see what it is.

Why is figure 6 a pie chart?? The components don’t add up to 100%, they are independent measures of each other.

295 and 296 should be below the figure.

Figure 7 – Why does (a) show the results of 409 predictions, while (e) shows 217. So it is 94% accurate, but it made half as many predictions? 83% vs 94% looks even more shaky now.

I am concerned about the experiment here - in terms of no cross-validation, and then differing-sized test datasets for the different approaches.

There is some poor grammar throughout.

Author Response

Response to Reviewer 2 Comments

Manuscript ID: applsci-1930763

Title: A Novel Deep Learning Approach for Deepfake Image Detection

Authors: Ali Raza*, Kashif Munir*, Mubarak Almutairi*

Dear Editor,

Thank you very much for allowing a resubmission of our manuscript, with an opportunity to address the reviewers’ comments.

We would like to thank the editor and all the worthy reviewers for their efforts, valuable comments and suggestions. The given comments improved the quality of our paper. Based on the feedback, we have extensively revised our manuscript. The updated manuscript with red highlighting indicates changes. The detailed modifications to address reviewers’ comments are provided point-by-point in the following. For clarity, we have marked our responses in blue.

REVIEWER 2

Point 1: This is an interesting area – and the overall approach is good. However, I think the presentation of the results could be improved, and I also think the actual analysis/experiment leaves room for improvement also – but perhaps some of this is from how it was presented in the paper.

Response 1: The authors are highly grateful for your generous efforts and insightful comments. We apologize for the inconvenience and inappropriate language that raised ambiguity. As per your valuable suggestions, the paper is extensively revised and improved to remove ambiguity.

Point 2: 54-64 feels clumsy – it is in two bullets – why? And then the contents don’t flow. I suggest just writing them as normal content that flows.

Response 2: We again apologize for the inconvenience caused. As per your valuable suggestions, we have written the contributions as normal content that flows. The updated manuscript from lines 53 to 64 follows as:

“The primary study contributions for deepfake detection are analyzed in detail. Transfer learning and deep learning-based neural network techniques are employed in comparison. The Xception, NAS-Net, Mobile Net, and VGG16 are the employed transfer learning techniques. A novel DFP approach based on a hybrid of VGG16 and convolutional neural network architecture is proposed for deepfake detection. The novel proposed DFP outperformed other state-of-the-art studies and transfer learning techniques. The hyperparameter tuning is applied to determine the best-fit parameters for the neural network techniques to achieve maximum performance accuracy scores. The training and validation effects of employed neural network techniques are examined in time-series graphs. The confusion matrix analysis of employed neural network techniques is conducted for performance metrics validation.”

Point 3: “Other studies have failed to achieve high metrics scores compared with our research study.”  This study is 94 – and there in the table has two 90s and a 93. Not convinced this is as significant a leap as you say to say that others have failed..

Response 3: We again apologize for the inconvenience and inappropriate language that raised ambiguity. As per your valuable suggestions, we have re-phrase the performance argument in the updated manuscript as:

“Our proposed study achieved high metrics scores for deepfake detection compared to previous studies.”

Point 4: There is no mention of cross-validation – so with such a small dataset, and then testing with just one lot of 10%, I am not convinced that 94% is actually an accurate number. The 10% testing set could be a ‘lucky’ 10%.

Response 4: As per your valuable suggestions, we have applied the k-fold cross-validations technique to our novel proposed model. The results of the cross-validations technique are added in the newly created subsection “K-Fold Cross Validation Results” follow as in the updated manuscript:

“The performance validation of our novel proposed approach is analyzed in Table ?. The k-fold cross-validation technique based on four folds of data is used for proposed model validations. The analysis shows that the proposed model achieved an average accuracy score of 95% with a standard deviation of 0.01. The cross-validation results demonstrate that our proposed model is in the generalized form to detect deepfakes from social media.”

We use the 90:10 splitting ratio to achieve high-performance scores for deepfake detection. In addition, the past published research articles which use a 90:10 splitting ratio for evaluating their performance results are the followings:

  • https://doi.org/10.1155/2021/4832864
  • https://doi.org/10.1186/s13040-017-0155-3
  • https://doi.org/10.1016/S0034-4257(00)00142-5
  • https://doi.org/10.1145/3447876
  • https://doi.org/10.1145/1015330.1015345

Point 5: “The François Chollet” – He is just a software engineer, not sure he warrants “The” as his title.

Response 5: As per your valuable suggestions, we have removed the line from the manuscript for clarity.

Point 6: “The Python programming tool [28] is used for evaluating all research experiments.” ??? what tool? Python is a language – so you mean the tool that is at reference 28. State it in the text, don’t make the reader go to the references to then see that it points to a paper, that they then need to go lookup to see what it is.

Response 6: We again apologize for the inconvenience and inappropriate language that raised ambiguity. As per your valuable suggestions, we have changed the term “tool” to “language” in the updated manuscript. Reference 28 uses the python programming language for conducting research experiments, which is why we have cited it in the text.

Point 7: Why is figure 6 a pie chart?? The components don’t add up to 100%, they are independent measures of each other.

Response 7: As per your valuable suggestions, we have removed the pie chart. We have re-design a bar chart to compare the loss score of applied neural network techniques. Figure 6 in the updated manuscript follows as:

Point 8: 295 and 296 should be below the figure.

Response 8: As per your valuable suggestions, we have updated lines 295 and 296 to re-set them below the figure.

Point 9: Figure 7 – Why does (a) show the results of 409 predictions, while (e) shows 217. So it is 94% accurate, but it made half as many predictions? 83% vs 94% looks even more shaky now.

Response 9: We again apologize for the inconvenience caused. The confusion matrix results only for the Xception model were misplaced unexpectedly during the first and second manuscript revisions. We have updated Figure 7(a) with the correct results. We have also reviewed and revised the other result scores for the Xception model. The updated manuscript containing Figure 7(a) as:

Point 10: I am concerned about the experiment here - in terms of no cross-validation, and then differing-sized test datasets for the different approaches.

Response 10: We again apologize for the inconvenience caused. We have applied the cross-validations techniques, adding the results in the updated manuscript. We have used the same test size for all models. Only the confusion matrix figure(a) was misplaced during revisions one and two. Now we have extensively revised the manuscript as per your suggestions

Point 11: There is some poor grammar throughout.

Response 11: We again apologize for the inconvenience and inappropriate language that raised ambiguity. The paper is extensively revised to remove ambiguity. We have addressed all the reviewer's comments with detailed justification and revised the manuscript.

Reviewer 3 Report (New Reviewer)

The paper is of good quality and supports materials of the highest caliber. The subject is also intriguing and timely. There are a few things that need to be modified, like a few typos and grammatical errors.

Author Response

Response to Reviewer 3 Comments

Manuscript ID: applsci-1930763

Title: A Novel Deep Learning Approach for Deepfake Image Detection

Authors: Ali Raza*, Kashif Munir*, Mubarak Almutairi*

Dear Editor,

Thank you very much for allowing a resubmission of our manuscript, with an opportunity to address the reviewers’ comments.

We would like to thank the editor and all the worthy reviewers for their efforts, valuable comments and suggestions. The given comments improved the quality of our paper. Based on the feedback, we have extensively revised our manuscript. The updated manuscript with red highlighting indicates changes. The detailed modifications to address reviewers’ comments are provided point-by-point in the following. For clarity, we have marked our responses in blue.

REVIEWER 3

Point: The paper is of good quality and supports materials of the highest caliber. The subject is also intriguing and timely. There are a few things that need to be modified, like a few typos and grammatical errors.

Response: The authors are highly grateful for your generous efforts and insightful comments. We apologize for the inconvenience and inappropriate language that raised ambiguity. As per your valuable suggestions, the paper is extensively revised to remove ambiguity.

Round 2

Reviewer 2 Report (New Reviewer)

Thanks for all the changes you have made - it has improved the paper. Although I have an outstanding issue - you have added the standard deviation - as +/- 0.01 - which I believe is incorrect. It is +/- 1. I think you have it as 1% and have transposed that to 0.01 - but your accuracy scores at 93-96 - and so it is 1 not 0.01.

Author Response

This manuscript is a resubmission of an earlier submission. The following is a list of the peer review reports and author responses from that submission.

Round 1

Reviewer 1 Report

his paper proposes a new deepfake image detection method. To explain the superiority of the new method, transfer learning and deep learning based image detection methods are studied and compared. The hyperparameter tuning is also performed to determine the best-fit parameters to achieve the maximum performance accuracy scores. Finally, tests are performed on time-series graphs and confusion matrix analysis is performed to validate performance metrics, illustrating that the novel proposed method outperforms state-of-the-art studies applied in the past.

The comments are as follows,

1. The paper describes a new method for deepfake image detection, but:

- The chapter on the new method needs to be expanded on, for example, explaining why three stacked convolutional neural network layers are chosen and what their advantages are,

- it has to be described why this method is better or different from other methods.

2. It should be explained why the authors chose the Mobile Net, VGG16, and NAS-Net Large for comparative experiments.

Author Response

The file is attached herewith for your reference.

Reviewer 2 Report

Ali Raza and coauthors proposed a novel Deep Fake Predictor (DFP) based on an optimized three-stacked convolutional neural network layer. The authors found that their DFP outperformed standard transfer learning techniques such as Mobile Net, VGG16, and NAS-Net with 99\% recall, 98\% accuracy and 98\% f1 scores for deepfake detection.

Overall, the authors delivered a well-structured manuscript. The introduction is a pleasant read for any general audience to quickly obtain a high-level overview of the deepfake-related literature. In the experiment and discussion section, the authors described all the parameters and setup in detail, including data acquisition/processing, model architectures, metrics, and hyperparameter tunings.

Nonetheless, I cannot overlook the fact that the authors provided little discussion to elaborate on why their DFP can outperform standard transfer learning methods. Besides presenting all the numbers about the experiment and results, the entire manuscript reads more like an experiment report rather than a research paper. 

Readers are not only interested in learning the fact that the proposed three-stacked CNN can achieve 98\% accuracy. More importantly, the authors need to provide enough insights to deepen our understanding of the topic so that it can guide future research. Please see the papers cited by the authors themselves in the literature review section for reference on how to provide research insights.

From what has been presented, it is hard to perceive and feel excited about its novelty and significance to deepfake research. 

In addition, the experiment results do not support the authors' claim that their proposed DFP outperforms transfer learning. VGG16, NAS-Net, and DFP are on par with each other considering a few percent experiment uncertainty. 

I feel that the paper does not meet the high standards and I cannot recommend the publication in its current form. 

Author Response

(The authors gave the same response as above.)

Round 2

Reviewer 2 Report

After a careful review of the updated manuscript, unfortunately, I still cannot recommend it for publication. 

Similar to the version 1, The updated manuscript still claims the main achievement is a novel Deep Fake Predictor (DFP), which is able to outperform standard transfer learning techniques with 99\% recall, 98\% accuracy. 

1. Lack of novelty

Despite the authors repeatedly argue that their DFP is novel, there is not enough evidence provided in the updated manuscript to demonstrate its "novelty".  In the updated manuscript, the authors' main argument for the novelty is that the proposed algorithm has much fewer parameters. But if we consider the dataset in this paper is very small (less than 2000 static image examples in total; several orders of magnitude smaller than other standard deep fake dataset), it is hardly convincing the fewer parameters in this proposal can be qualified as "novelty".

Furthermore, one must compare its DFP to other state-of-art DFPs on the same dataset to claim the novelty of fewer parameters. In the manuscript, the authors compared their proposal to models that are never intended or optimized for deep fake detections. Although this comparison itself can be interesting, yet it does not support authors' claim that their proposed DFP is novel and scientifically significant. The authors should consider to compare performance on the same dataset against DFPs reported in the literature such as those those cited in the table 1 of this manuscript.

2. no evidence to support performance claim

More importantly, the experiment results still do not support the authors' claim that the proposed DFP can outperform standard transfer learning techniques, which the authors also agree in their response (see response 6 from the authors).

If we compare the proposed DFP to transfer learning models in figure 5, it is clear that 

1) accuracy: proposed DFP is 98% against VGG16's 97%

2) recall: proposed DFP is 99% against mobile Net's 98%

3) F1: proposed DFP is 98% against VGG16's 97%

1% difference can easily come from the specific distribution in the data or training process, which cannot be justified as a statistically significant gain. 

The study (https://doi.org/10.3390/e21111078) mentioned by the author is not on the same dataset as this manuscript. I failed to see any relevance to our discussion here. 

Suggestions:

to proof the novelty of your proposed DFP, the authors must provide a detailed and scientific sound argument to demonstrate how exactly their proposal 1) leverages information that was not studied before or 2) invents a new algorithm that is scientifically unique

In the literature cited by the authors, a series of novel deep fake studies have been reported. For example, 1) the first-time using YOLO to extract faces from videos, 2) first-time using ensemble learning 3) first-time leveraging eye-blinking information 4) first-time using spectral, spatial and temporal inconsistencies information...